# Evaluation of Parameters for Estimating the Postmortem Interval of Skeletal Remains Using Bovine Femurs: A Pilot Study

**DOI:** 10.3390/diagnostics10121066

**Published:** 2020-12-09

**Authors:** Midori Nagai, Koichi Sakurada, Kazuhiko Imaizumi, Yoshinori Ogawa, Motohiro Uo, Takeshi Funakoshi, Koichi Uemura

**Affiliations:** 1Department of Forensic Medicine, Tokyo Medical and Dental University, 1-5-45 Yushima, Bunkyo-ku, Tokyo 113-8519, Japan; naglegm@tmd.ac.jp (M.N.); funakoshi.legm@tmd.ac.jp (T.F.); kuemura.legm@tmd.ac.jp (K.U.); 2Department of Forensic Dentistry, Tokyo Medical and Dental University, 1-5-45 Yushima, Bunkyo-ku, Tokyo 113-8510, Japan; 3Second Biology Section, First Department of Forensic Science, National Research Institute of Police Science, 6-3-1 Kashiwanoha, Kashiwa, Chiba 277-0882, Japan; imaizumi@nrips.go.jp (K.I.); y-ogawa@nrips.go.jp (Y.O.); 4Department of Advanced Biomaterials, Tokyo Medical and Dental University, 1-5-45 Yushima, Bunkyo-ku, Tokyo 113-8510, Japan; uo.abm@tmd.ac.jp

**Keywords:** bone density, DNA degradation, element concentration, postmortem interval, skeletal remains, inductively coupled plasma optical emission spectrometry

## Abstract

The postmortem interval (PMI) of victims is a key parameter in criminal investigations. However, effective methods for estimating the PMI of skeletal remains have not been established because it is determined by various factors, including environmental conditions. To identify effective parameters for estimating the PMI of skeletal remains, we investigated the change in bone focusing on the amount of DNA, element concentrations, and bone density that occurred in the bone samples of bovine femurs, each maintained under one of five simulated environmental conditions (seawater, freshwater, underground, outdoors, and indoors) for 1 year. The amount of extracted mitochondrial DNA (mtDNA; 404 bp fragment) decreased over time, and significant DNA degradation (*p* < 0.01), as estimated by a comparison with amplification results for a shorter fragment (128 bp), was detected between 1 month and 3 months. Eleven of 30 elements were detected in samples by inductively coupled plasma optical emission spectrometry, and Na and Ba showed significant quantitative differences in terms of environmental conditions and time (*p* < 0.01). This preliminary study suggests that the level of DNA degradation determined by real-time polymerase chain reaction and element concentrations determined by inductively coupled plasma optical emission may be useful indices for estimating the PMI of victims under a wide range of environmental conditions. However, this study is a limited experimental research and not applicable to forensic cases as it is. Further studies of human bone with longer observation periods are required to verify these findings and to establish effective methods for PMI estimation.

## 1. Introduction

In forensic science, the postmortem interval (PMI) of victims is one of the most important factors for criminal investigations. In particular, in cases of skeletal remains, the PMI becomes very difficult to estimate because soft tissues decrease substantially or completely. Many researchers have focused on the development of novel methods, including infrared (IR)/Raman microscopic imaging techniques [1,2], inductively coupled plasma optical emission spectroscopy (ICP-OES) [3], or microcomputed tomography (µCT) [4,5]. In addition, combinations of techniques have been reported, such as the combined use of different physicochemical methods [6], luminol and microscopic techniques to screen bones before the use of radiocarbon dating [7], ultraviolet (UV)-induced fluorescence and 490 nm induced fluorescence [8], and µCT, mid-infrared microscopic imaging, and energy-dispersive X-ray mapping [9]. Molecular biology approaches have been used to evaluate the relationships between environmental insults and DNA degradation [10,11,12,13]. Because DNA is often highly fragmented [14,15], mitochondrial DNA could be more useful than nuclear DNA [15,16].

Element analyses are often used in the field of archeology and have recently been applied in forensic science to examine the degree of bone degradation. Walden et al. used ICP-OES to measure bone degradation [3]. Gallelo et al. [17] compared elemental compositions between bones and the environment. Because elements in compact bone are not easily affected by the environment when the bone density is high, elemental compositions and bone density should be measured at the same time. Considering the structure of compact bone, rich in narrow ducts, it is necessary to establish alternative methods for obtaining the precise density. µCT is a non-destructive method used to observe the detailed three-dimensional structure of small specimens. After analyses by µCT, samples can be used for other tests, such as DNA or elemental analyses. When only a small bone fragment is available, µCT might be useful. We applied this method to precisely measure the density of compact bone with the aim of establishing an effective indicator of the PMI.

Changes in bone exposed to various environmental conditions have not been evaluated from the viewpoints of biological, chemical, and physical properties over long periods of time. Therefore, in this study, the changes in bone were investigated to find effective parameters for estimating the PMI of skeletal remains on the basis of analyses of DNA quantity, element concentrations, and bone density in bovine compact bones, each exposed to one of five environmental conditions (i.e., seawater, freshwater, underground, outdoor, and indoor conditions) for 1 year.

## 2. Materials and Methods

### 2.1. Bone Sample Preparation

A femur was taken from a 30 month old cow. After removing soft tissues, including the periosteum, the bone shaft was cut into two parts along the bone axis using an electric cast cutter (Stryker Corporation, Kalamazoo, MI, USA) and the bone marrow was removed. Bone shafts were divided into eight parts by vertical and horizontal cuts. These parts were then cut into pieces weighing approximately 1 g each. All of these bone pieces included both the periosteum and the bone marrow sides. Fan-shaped bone pieces were obtained, and two horizontally cut surfaces were smoothed by slicing off their thin surfaces with a precision cutting machine (aqra PRECISO CL40; JEOL Ltd., Tokyo, Japan) and diamond disc (Horico Dental Hopf, Ringleb & Co. GmbH & Cie, Berlin, Germany) with cooling water. The specimens were washed with neutral detergent and Milli-Q ultrapure water (Merck KGaA, Darmstadt, Germany). After drying at room temperature, they were stored at −30 °C until use. The specimens were analyzed almost a week after they were removed from the investigated conditions. The control was analyzed at the same time after 3 days.

### 2.2. Environmental Conditions

Five environmental conditions were simulated—immersion in seawater, immersion in freshwater, buried underground, outside, and indoors—considering the conditions in the area surrounding Tokyo, Japan, particularly soil type. The seawater was prepared by dissolving powdered artificial seawater (Nihon Kaisui, Tokyo, Japan) in distilled water. For the freshwater treatment, Mount Fuji mineral water was used (Suntory, Osaka, Japan). Each specimen was immersed in 50 mL of liquid in a disposable test tube. These liquids were exchanged once per week throughout the experimental period to keep them as fresh as possible. Most of the soil in the Tokyo area is a mixture of red and black soil. Therefore, we mixed non-heat-treated red and black soil in a 1:1 ratio for the underground condition. Soil type is a key parameter in forensic analyses. The black soil was a Japan-specific soil called “Kurobokudo”, which results from field burning, volcanic ash, and humus [18]. For outdoor conditions, an airy site with a roof in Tokyo was used. The temperature in Tokyo was 16.8 °C, on average, and was highest (39 °C) in July in the summer and lowest (−4 °C) in January in the winter in 2018. To simulate indoor conditions, specimens were separately placed in a laboratory with a temperature of 25 °C. The water immersion and soil burial conditions were also simulated in this room. Bone samples prepared as described above were used as controls. Five bone specimens were each placed in a 50 mL tube and exposed to the conditions for various durations (3 days, 1 week, 1 month, 3 months, 6 months, and 1 year). The tubes were placed upright on a tube rack and covered with a plastic film to avoid contamination with insects and dust. This treatment also kept soils moist throughout the experiment. Overall, samples from five environmental conditions (seawater, freshwater, underground, outside, and indoors) at six time points (3 days, 1 week, 1 month, 3 months, 6 months, and 1 year) and a control were investigated. We prepared two sets of specimens for each environmental condition (one for DNA quantitation and elemental analysis, and the other for density analysis). All experiments were started at the same time in January 2018 and ended in January 2019.

Collected specimens were washed with a neutral detergent using a sponge, followed by several washes with ultrapure water. After drying samples under a plastic hood at room temperature for 24 h, parts for DNA analyses were pulverized using a Multi-beads shocker (MB1200; Yasui Kikai Co. Ltd., Osaka, Japan) at 2000 rpm for 20 s and at 3000 rpm for 10 s or 20 s until pulverized completely. After 24 h of drying at room temperature, these pulverized samples were kept at −30 °C until use. The specimens were analyzed almost a week after they were removed from the investigated conditions. The control was analyzed at the same time after 3 days.

### 2.3. DNA Extraction and Quantification

Approximately 0.2 g of bone powder was transferred into 50 mL tubes. Decalcification was performed using 30 mL of 0.5 M ethylenediaminetetraacetic acid (EDTA) at 56 °C for 24 h. To remove EDTA, the powder was washed with ultrapure water five times by centrifugation at 3000 rpm for 5 min. The powder was digested with 25 µL of Proteinase K (20 mg/mL; Takara Bio Inc., Shiga, Japan) and 1000 µL of Proteinase K buffer (Takara Bio Inc.; http://catalog.takara-bio.co.jp/PDFS/9034_DS_j.pdf) at 56 °C for 24 h. Proteinase K buffer contained 0.01 M Tris-HCl (pH 7.8), 0.01 M EDTA, and 0.5% sodium dodecyl sulfate. The tube was not shaken during the processing of EDTA and Proteinase K. DNA was extracted using the QIAamp DNA Mini Kit (Qiagen Benelux B.V., Venlo, Netherlands) according to the manufacturer’s instructions. DNA extracts were adjusted to 50 µL with buffer AE (Qiagen Benelux B.V.).

Two primer sets targeting the 16S ribosomal RNA (rRNA) region of bovine mitochondrial DNA (mtDNA) were used (Table 1). The target products were 128 bp [19] and 404 bp, and the degree of DNA degradation was estimated by comparing amplification. The primers for the longer target were designed using Primer 3 Plus (https://primer3plus.com/), and the specificity was confirmed using Primer-BLAST (https://www.ncbi.nlm.nih.gov/tools/primer-blast/). Real-time PCR amplification was performed using a Thermal Cycler Dice Real Time System III (Takara Bio Inc.) according to the manufacturer’s instructions. TB Green Premix Ex Taq II Tli RNaseH Plus (Takara Bio Inc.) was used for amplification according to the manufacturer’s instruction. The total volume of PCR reaction was 25 µL and the template DNA was 2 µL. For the 128 bp target, reaction conditions were 95 °C for 30 s, followed by 40 cycles at 95 °C for 5 s and 60 °C for 30 s. For the 404 bp target, cycling conditions were set to 95 °C for 30 s, followed by 40 cycles at 95 °C for 5 s and 60 °C for 40 s. The extraction blank control was the negative control for each condition and PCR blanks were also tested. DNA extracts were not diluted. Samples were quantified in triplicate. To generate a calibration curve, serial dilutions of the bovine mtDNA positive control (Promega KK, Madison, WI, USA: Code No. MO-COW 15000/Lot No. PCW180818) were used. Total DNA quantitation was performed using the NanoDrop One Spectrophotometer (ThermoFisher Scientific, Waltham, MA, USA) with 2 µL of each extracted DNA solution according to the manufacturer’s instructions.

### 2.4. Elemental Analysis by ICP-OES

Qualitative and semiquantitative analyses of 30 elements were performed as a preliminary test to select appropriate elements for the study. Bone powder obtained from the control, seawater, freshwater, and underground specimens after 6 months (*n* = 1) was weighed, and 0.1 g of each sample was hydrolyzed with nitric acid. After drying, pyrolysis was performed with perchloric acid and nitric acid. After drying to leave several microliters, sample solutions were prepared immediately by the addition of 20 mL of diluted nitric acid. The elements Li, Be, B, Na, Mg, Al, Si, P, K, Ca, Ti, V, Cr, Mn, Fe, Co, Ni, Cu, Zn, As, Se, Sr, Zr, Mo, Ag, Cd, Sn, Sb, Ba, and Pb were semi-quantitated by the single inspection method with the standard solution (Multi-Element Calibration Standard 3,4, and 5, PerkinElmer, Inc., Waltham, MA, USA and Custom Assurance Standard XSTC-22, SPEX CertiPrep, Metuchen, NJ, USA) using ICPS-8100 (Shimadzu Corporation, Kyoto, Japan).

On the basis of this preliminary analysis, elements were chosen for precise quantitation. Specimens from the seawater, freshwater, and underground conditions after 6 months and 1 year, as well as the control (*n* = 3), were used for further quantitative analyses, following the protocol used for the preliminary test. The concentrated solutions obtained from specimens were diluted with 20 mL of diluted nitric acid and applied to the analyses. The calibration curve was prepared from serially diluted standard solutions of Na, Zn, and Ba (Kanto Chemical Co., Inc., Tokyo, Japan). The elemental analysis by ICP-OES was supported by the analysis service (Shimadzu Corporation, Kyoto, Japan).

### 2.5. Bone Density Measurement by Micro X-Ray CT

Control specimens (*n* = 5) and the specimens obtained from seawater, freshwater, underground, outdoor, and indoor conditions at the periods of 6 months and 1 year (*n* = 5 each) were scanned with the micro X-ray CT system (InspeXio SMX-225CT FPD FR; Shimadzu Corporation). Scanning parameters for cone-beam CT were 200 kV and 70 µA, with the default rotation and collection frequency optimized for observing light metals. The slice thickness was 0.020 mm. The scanned data were analyzed using VGStudio Max 3.2 (Volume Graphics GmbH, Heidelberg, Germany) by three-dimensional volume rendering methods, and volumes in mm^3^ were obtained. All procedures were performed by a single researcher, and the samples were analyzed in random order. Bone density was calculated on the basis of the volume and mass measured before the scan.

### 2.6. Statistical Analysis

Statistical analyses were conducted using Statcel (The Useful add-in Forms on Excel, 4th ed.). The interquartile range (IQR) was checked and outliers were omitted before statistical analyses. To check normality and homoscedasticity, *F*-tests were performed. Student’s *t*-test, one-way analysis of variance with Turkey–Kramer post hoc tests, and Dunnett’s test were used. The level of significance was set to 1%.

## 3. Results

### 3.1. Visual Observation of Bone Samples in Five Environmental Conditions

Figure 1 shows images of the specimens subjected to conditions for 1 year. They were photographed after washing and drying. The control was yellowish-white (Figure 1a). The colors of the specimens in seawater (Figure 1b) and freshwater (Figure 1c) were nearly the same as that of the control, while the specimens in outdoor and indoor conditions (Figure 1d,e) were whitish. The underground specimens (Figure 1f,g) showed variable coloration, including brownish with many dark brown spots and wholly light brown.

### 3.2. DNA Quantitation of Bones Exposed to Various Environmental Conditions

DNA quantitation results obtained by real-time PCR for the 404 bp target are summarized in Figure 2. DNA quantity decreased over time in most conditions, except indoor conditions. DNA quantities for specimens in seawater and freshwater at 1 week were significantly lower than that of the control (*p* < 0.01). The DNA quantity of the underground specimen became significantly lower than that of the control at just 3 days (*p* < 0.01). DNA quantities for indoor specimens did not decrease significantly over a 1 year period and were significantly higher than that of the control at 3 days and 1 month (*p* < 0.01). Comparing environmental conditions, although decreases in DNA quantities occurred the earliest in underground conditions among the four conditions except indoor, they showed the same extent of decrease at 1 year.

Figure 3 shows differences in DNA quantities among five conditions at 1 month and 1 year. DNA quantities were significantly lower (*p* < 0.01) for the specimens subjected to freshwater and underground conditions and significantly higher for indoor conditions (*p* < 0.01) than for the control at 1 month. DNA quantities were significantly lower (*p* < 0.01) compared with the control at 1 year under all conditions. The DNA quantity for samples in seawater, outdoor, and indoor conditions decreased between 1 month and 1 year.

Figure 4 summarizes the degree of DNA degradation over time, as estimated by the difference in amplification of the two PCR targets (404 bp and 128 bp) by real-time PCR. Although the decreases in DNA quantities showed similar trends for the short (128 bp) and long (404 bp) targets in all five environmental conditions, DNA quantities were significantly higher for the long target at 3 days in seawater (*p* < 0.05), and similar trends were seen at 3 days in freshwater and indoor conditions. Moreover, DNA quantities were significantly lower for the long target at 3 months in seawater (*p* < 0.05), as well as 1 month and 3 months in underground and outdoor conditions (*p* < 0.05 or *p* < 0.01). Significant differences between conditions were not observed after 6 months.

Figure 5 summarizes the results for total DNA quantitation using a spectrophotometer. DNA quantities from specimens in seawater, freshwater, and underground conditions were significantly lower than that in the control (*p* < 0.01) at 3 months, 1 week, and 3 days, respectively. For samples in outdoor conditions, DNA quantity did not change significantly over time. In indoor conditions, the DNA quantity was significantly higher than that in the control (*p* < 0.01) at 3 days and 6 months, and it was significantly lower than that in the control (*p* < 0.05) at 1 year. Compared with real-time PCR, spectrophotometer showed higher DNA quantities (for example, in an underground sample, real-time PCR: 0.18 ± 0.06 ng/μL; spectrophotometer: 23.95 ± 6.31 ng/μL).

### 3.3. Elemental Analysis by ICP-OES

Qualitative and semiquantitative analyses of 30 elements in pulverized specimens were performed by ICP-OES. Analyzed samples were the control, seawater, freshwater, and underground specimens after 6 months (*n* = 1). In a preliminary test, 11 of 30 elements were detected by comparisons with standard solutions (Table 2). Among these, Na, Zn, and Ba showed variations among conditions. As such, these three elements were quantitatively analyzed in the specimens subjected to seawater, freshwater, and underground conditions (6 months and 1 year, *n* = 3 each), as well as in control samples. The results are summarized in Figure 6. For specimens at 6 months and 1 year in freshwater conditions, the Na content was significantly lower than that in the control (*p* < 0.01). The Na content did not differ between 6 months and 1 year. Zn did not show a significant change in any conditions. Ba showed a significant decrease in all environments, except underground, at 6 months (*p* < 0.01). In underground conditions, there was a significant difference in the Ba content between 6 months and 1 year (*p* < 0.05), but no difference was found in the specimens maintained in seawater and freshwater.

### 3.4. Bone Density

In all conditions at 6 months and 1 year, micro X-ray CT (*n* = 5) showed the lamella structure and many small canals in the compact bone that are frequently observed in younger animals (Figure 7). The volume of each bone specimen was obtained from three-dimensional images, and bone density was calculated from the mass (Table 3). One sample for outdoor conditions at 1 year (Sample 1) was identified as an outlier and was, therefore, omitted from statistical analyses. Although the densities tended to decrease at 6 months and increase at 1 year, these changes were not statistically significant.

## 4. Discussion

Estimating the PMI of skeletal remains a major challenge in forensic science, with various techniques previously used, including IR/Raman microscopic imaging, ICP-OES, and radiocarbon dating [1,2,3,7]. Although these studies have provided valuable information, a reliable marker has not been established to estimate the PMI of skeletal remains. In the present study, we searched for effective indices for the accurate estimation of the PMI of bones. As a preliminary study, we evaluated the postmortem modifications after skeletonization on the basis of removing soft tissues and exposing the internal zone of bone. We investigated the change of bone as a function of DNA quantity, elemental composition, and bone density in bovine femurs experimentally exposed to five simulated conditions (seawater, freshwater, underground, outdoors, and indoors) for a maximum of 1 year. The DNA quantity decreased substantially in underground conditions after 3 days. The adsorption of DNA by soil makes extraction difficult, and this is a particular issue in the black soil unique to Japan [18]. In addition, PCR inhibitors in soil can affect the DNA quantity that measured by real-time PCR [20,21,22]. Furthermore, the quantity was significantly lower in the specimens maintained in freshwater and seawater. Cartozzo et al. [23] suggested that the greatest threat to DNA in bones submerged in water may be strand breakage resulting from hydrolysis (deamination, depurination, and depyrimidination). The decrease in DNA observed in this study may be due to this insult.

In addition to the PCR results for the 404 bp product, we amplified a 128 bp target to estimate the degree of DNA degradation. Lower amplification in the 404 bp target was observed at the maximum of 3 months. This trend suggests that the real-time PCRs targeting differently sized targets could be an effective index for DNA degradation of short terms. As a factor contributing to degradation, the process for decalcification should be considered. We decalcified specimens by incubation with EDTA. Incubation protocols vary among studies [24,25]. We selected 24 h at 56 °C; relatively warm and long incubation conditions might affect DNA quality. Total DNA varied in amount with conditions and exposure periods. In this quantitation, exogenous DNA originated from microorganisms proliferating in bone and endogenous DNA of bone itself is quantitated at once regardless of the degree of DNA degradation. Exogenous DNA begins to increase in parallel with severe bone degradation over a long exposure period. The values of total DNA, therefore, depend on the valance between the amount of endogenous and exogenous DNA. Variations in increase and decrease observed in this short-term study must have been derived from this characteristic of quantitation. However, if an increase in total DNA is clearly observed in more highly degraded specimens, this quantitation can be used as an index for bone degradation. In addition, Sessa et al. [26] reported the influence of the insects in postmortem changes on bones. They mentioned that the presence of insects feeding on the marrow could be one of the reasons for the poor DNA quality; however, other factors such as the environmental conditions where the skeleton was found cannot be excluded. The results may support ours.

An ICP-OES analysis was performed to find effective elements for estimating the PMI of bone. In qualitative and semiquantitative analyses, Na, Zn, and Ba showed variation in concentration among the conditions. Thus, some elements can be used to estimate the PMI as shown in the previous report [3]. However, additional analyses with more time points at the initial stage are needed to clarify the relation between elements and the PMI.

No clear change in bone density was observed in this study. Longato et al. [9] suggested that specimens with a short PMI have higher bone densities compared with specimens with a long PMI according to a study using anthropological human bones. Delannoy et al. [27] suggested that there are significant differences in human bone mass loss between environments (indoors, soil indoors, soil outdoors, soil outdoors protected from rain). In the present study, we used bovine femurs. Aerssens et al. [28] reported that bone density varies depending on the type and part of bone, as well as among individuals. Differences in bone properties might explain differences in results between studies.

In the present study, we investigated various parameters for estimating the PMI of skeletal remains. We confirmed the usefulness of real-time PCR for determining the level of DNA degradation. In particular, the difference in amplification of the two PCR targets (404 bp and 128 bp) may be an effective index for estimating the PMI after 1–3 months. On the other hand, element concentrations determined by ICP-OES, particularly the concentrations of Na and Ba, changed with respect to environmental conditions and time. However, additional analyses with more time points are needed to clarify the relationship between the elements and the PMI.

We admit that this study is a limited experimental research and not applicable to forensic cases as it is. In particular, we removed soft tissues prior to environmental exposure, which is not realistic in actual cases. Moreover, we did not perform a blind PMI evaluation according to the data obtained in this study. However, it might be said that this study is valuable for identifying the changes in bone after skeletonization. Further studies using more realistic specimens would bring more informative results for estimating the PMI. Regarding the result that bone density remained nearly unchanged in this study, it should be reinvestigated using pig or dog femurs, which are more similar to those of human.

## Figures and Tables

**Figure 1 diagnostics-10-01066-f001:**
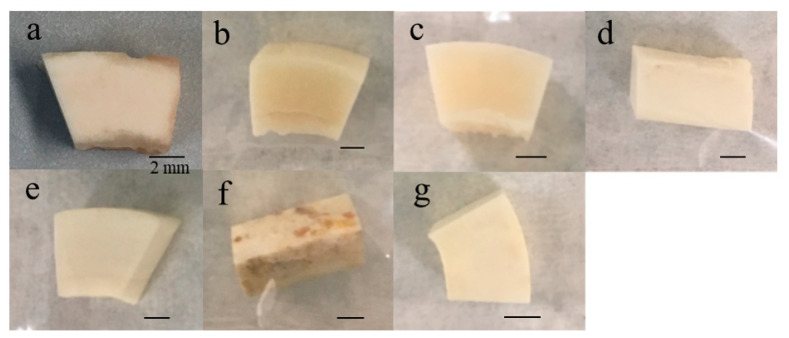
Bone samples maintained under five environmental conditions for 1 year: (**a**) control, (**b**) seawater, (**c**) freshwater, (**d**) outdoors, (**e**) indoors, and (**f**,**g**) underground.

**Figure 2 diagnostics-10-01066-f002:**
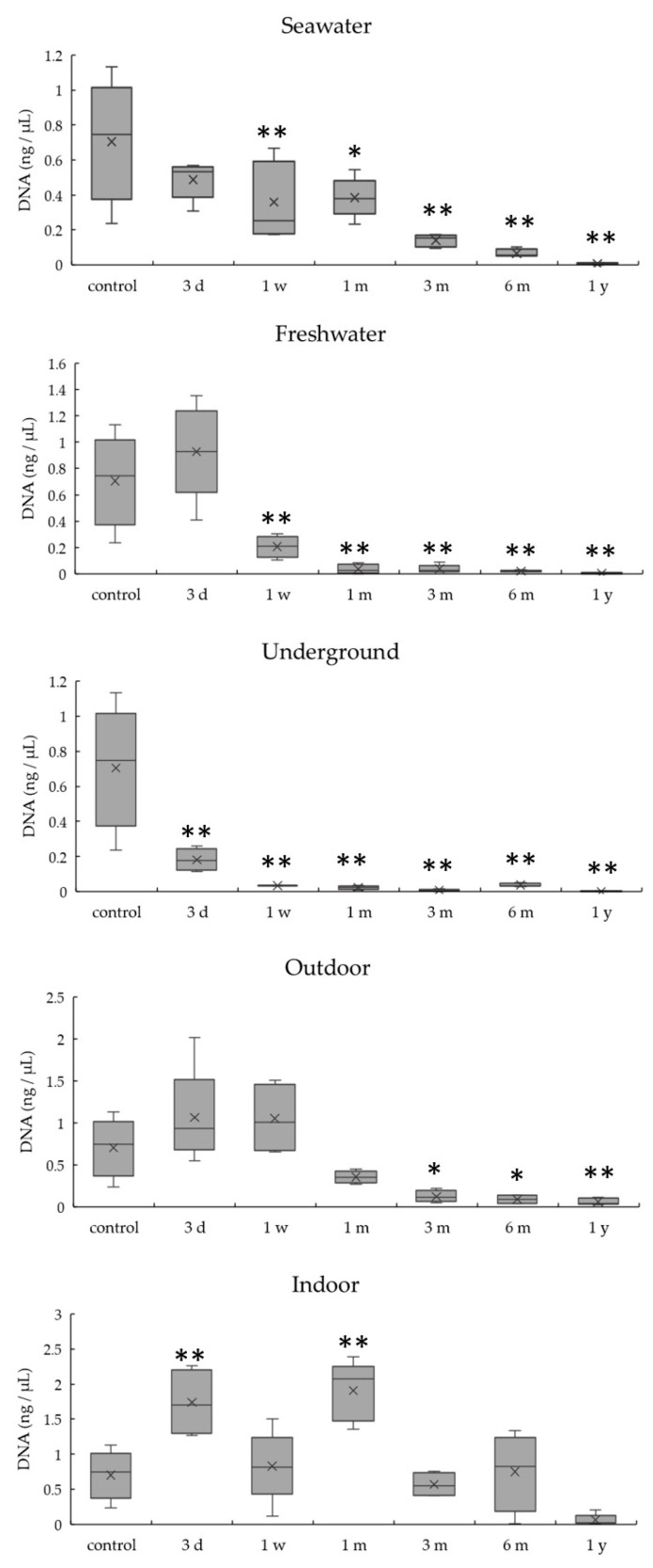
DNA quantitation of a 404 bp target by real-time PCR in bones exposed to five environmental conditions for 1 year. Data are presented as means and standard deviations. * *p* < 0.05 vs. control, ** *p* < 0.01 vs. control.

**Figure 3 diagnostics-10-01066-f003:**
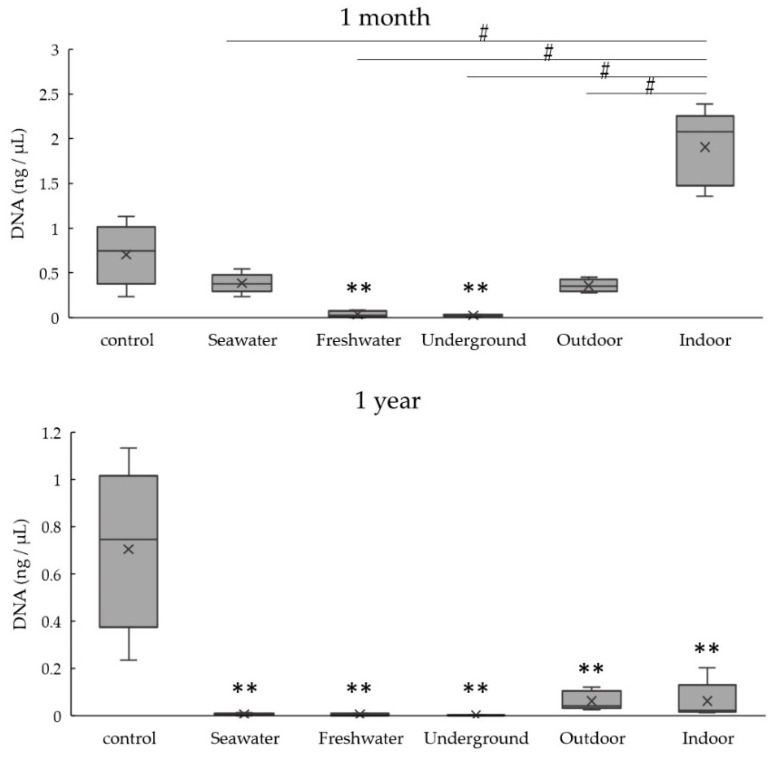
Differences in the amount of DNA between the five environmental conditions at 1 month and 1 year. Data are presented as means and standard deviations. ** *p* < 0.01 vs. control, ^#^
*p* < 0.01.

**Figure 4 diagnostics-10-01066-f004:**
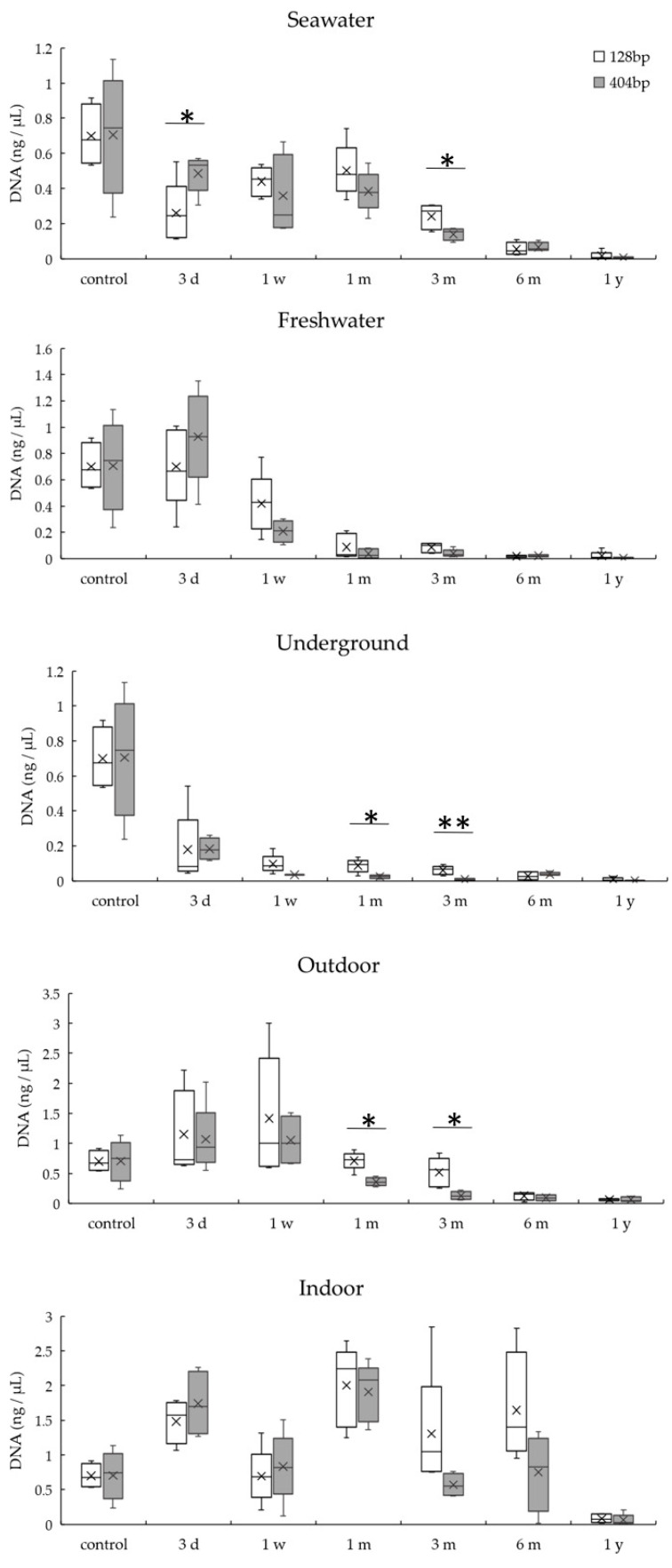
DNA degradation over time based on the amplification of 404 bp and 128 bp fragments by real-time PCR (404 bp vs. 128 bp). * *p* < 0.05, ** *p* < 0.01.

**Figure 5 diagnostics-10-01066-f005:**
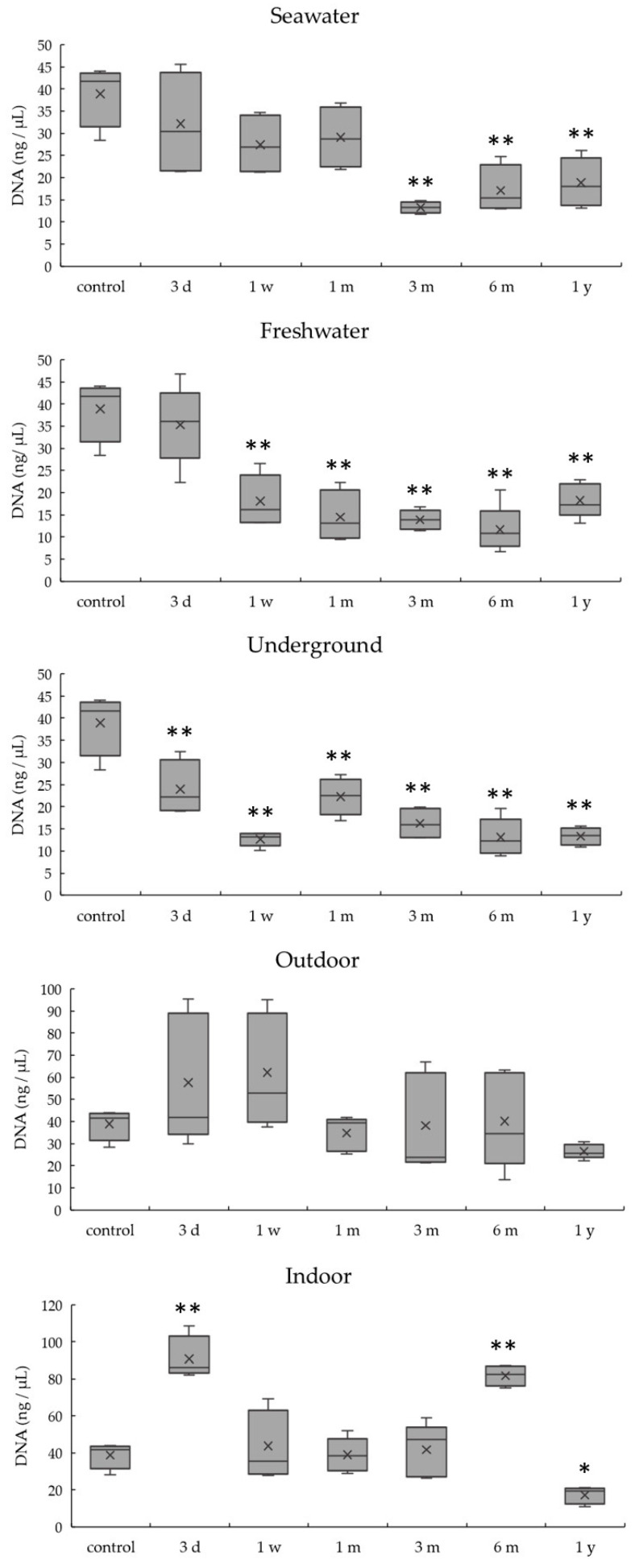
DNA quantitation by the absorbance method under five environmental conditions for 1 year. Means and standard deviations are presented. * *p* < 0.05 vs. control, ** *p* < 0.01 vs. control.

**Figure 6 diagnostics-10-01066-f006:**
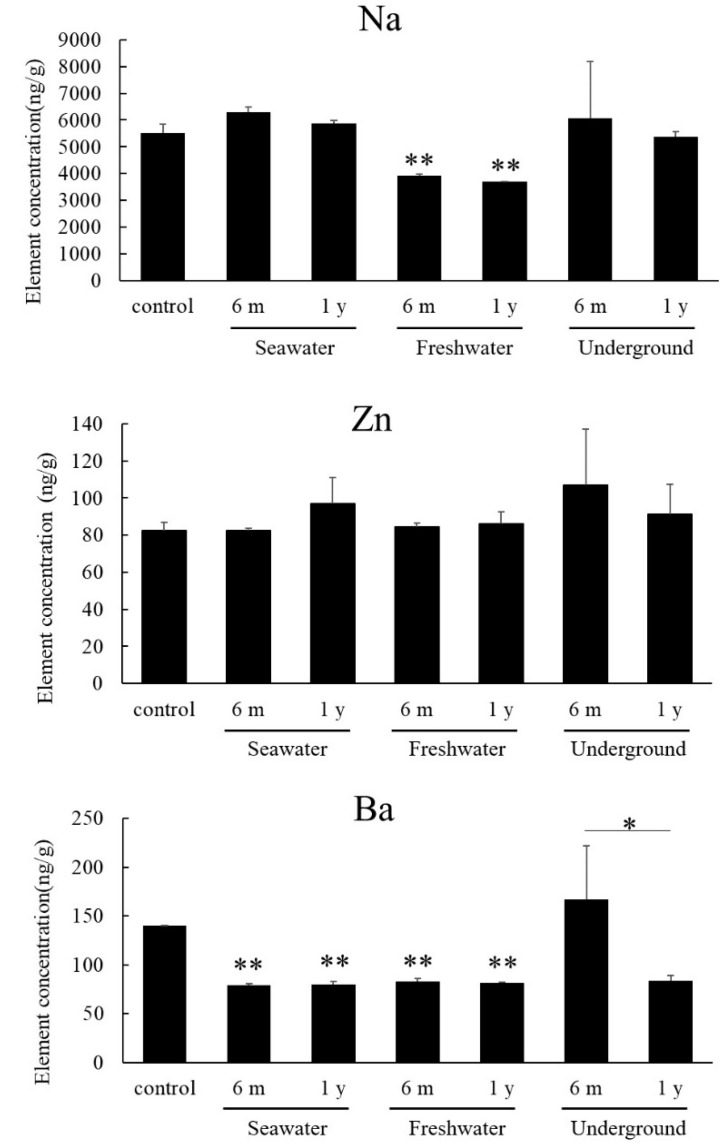
Quantitative analysis of Na, Zn, and Ba in the control and three environmental conditions by inductively coupled plasma optical emission spectroscopy (ICP-OES). Means and standard deviations are presented. * *p* < 0.05, ** *p* < 0.01 vs. control.

**Figure 7 diagnostics-10-01066-f007:**
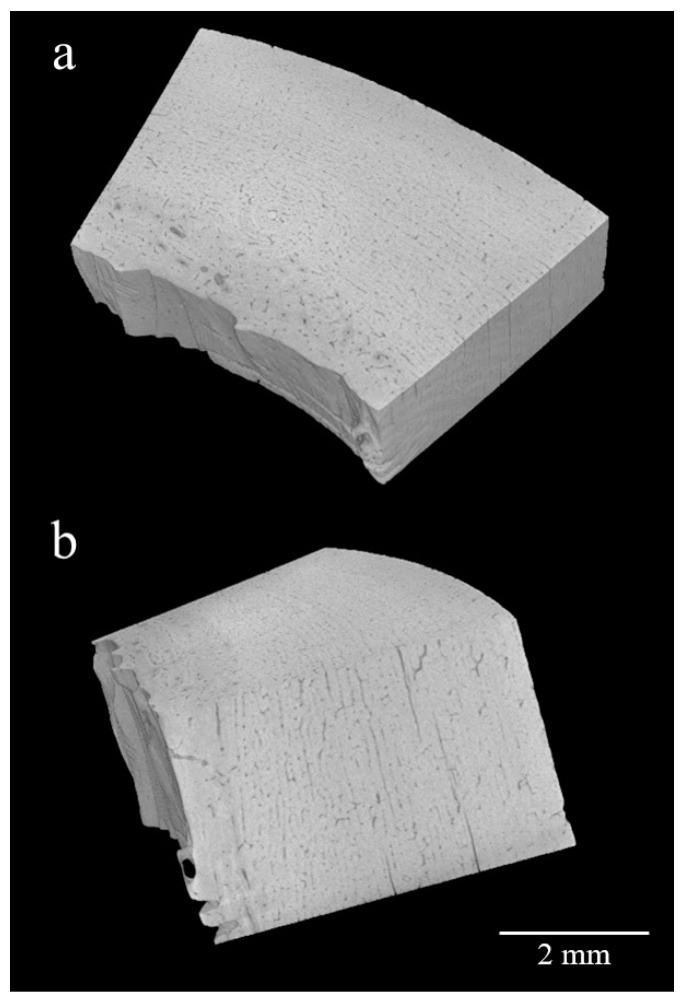
Three-dimensional (3D) images obtained by X-ray microcomputed tomography. The sample was maintained in freshwater for 1 year: (**a**) longitudinal direction; (**b**) minor axis direction.

**Table 1 diagnostics-10-01066-t001:** Primers designed in this study.

PCR Product Size (bp)	Primer Sequences (5’–3’)	
	Forward	Reverse
404	CTTGTATGAATGGCCGCACG	TGATGGTGCAACCGCTATCA
128	AACCATTAAGGAATAACAACAA	AAATCACTCTATCGCTCATTG

**Table 2 diagnostics-10-01066-t002:** Qualitative and semiquantitative analyses of 30 elements by ICP-OES.

Element	Estimated Concentration				Detection Limit
	Control	Seawater	Freshwater	Underground	
	(μg/g)	(μg/g)	(μg/g)	(μg/g)	(μg/g)
Li	ND *	ND	ND	ND	10
Be	ND	ND	ND	ND	0.2
B	ND	60	ND	ND	20
Na	5000	7000	4000	6000	100
Mg	5000	5000	4000	5000	2
Al	ND	ND	ND	ND	10
Si	ND	ND	ND	ND	20
P	100,000	100,000	100,000	100,000	20
K	ND	ND	ND	ND	400
Ca	200,000	200,000	200,000	200,000	200
Ti	ND	ND	ND	ND	40
V	ND	ND	ND	ND	4
Cr	20	ND	ND	5	4
Mn	3	ND	ND	ND	1
Fe	300	ND	ND	ND	20
Co	ND	ND	ND	ND	10
Ni	ND	ND	ND	ND	10
Cu	ND	ND	ND	ND	10
Zn	60	70	90	60	10
As	ND	ND	ND	ND	40
Se	ND	ND	ND	ND	40
Sr	200	200	100	200	0.2
Zr	ND	ND	ND	ND	4
Mo	ND	ND	ND	ND	10
Ag	ND	ND	ND	ND	10
Cd	ND	ND	ND	ND	4
Sn	ND	ND	ND	ND	20
Sb	ND	ND	ND	ND	20
Ba	100	70	80	100	2
Pb	ND	ND	ND	ND	40

* ND: not detected.

**Table 3 diagnostics-10-01066-t003:** Bone density determined by X-ray microcomputed tomography.

Sample No.		Seawater		Freshwater		Underground		Outdoors		Indoors	
	Control	6 m *	1 y ^†^	6 m	1 y	6 m	1 y	6 m	1 y	6 m	1 y
Sample 1	1.864	1.972	2.060	2.026	1.997	1.934	1.981	1.995	2.444	2.005	2.040
Sample 2	1.717	1.929	2.020	1.965	2.079	1.961	2.042	1.984	1.985	2.018	2.054
Sample 3	2.089	2.006	2.089	2.056	2.029	2.010	2.021	2.041	2.057	2.053	2.035
Sample 4	2.162	2.027	2.048	2.027	2.072	2.002	2.003	2.003	2.004	2.045	2.026
Sample 5	2.542	1.981	2.045	2.027	2.056	1.973	1.988	1.996	1.995	2.008	2.062
Average	2.075	1.983	2.053	2.020	2.047	1.976	2.007	2.004	2.097	2.026	2.043
SD	0.316	0.037	0.025	0.033	0.034	0.031	0.025	0.022	0.196	0.022	0.015

* 6 m: 6 months; ^†^ 1 y: 1 year (mg/mm^3^).

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
