# Peer review of "Evaluation of Parameters for Estimating the Postmortem Interval of Skeletal Remains Using Bovine Femurs: A Pilot Study"

_diagnostics, 2020, doi:10.3390/diagnostics10121066_

Round 1

Reviewer 1 Report

The paper is linked, as the authors write, to verifying changes in bone exposed to various environmental conditions over long periods of time. The authors estimating the PMI of skeletal remains based on analyzes of DNA quantity, element concentrations, and bone density in one bovine compact bones, each exposed to one of five environmental conditions (ie, seawater, freshwater, underground, outdoor, and indoor conditions) for 1 year.
The paper has seen an improvement work compared to the previous version and with further modifications it may become acceptable.
The major limitations that still persist in the study are related to the lack of description of the objective limits of this experiment.
Furthermore, the minimum procedures for evaluating the results have not been respected, which should include an accurate methodology explaining who does what and on which scientific basis the estimation of the differences is reached.
Finally, I did not understand which PMI interval range is transferable from the experimental study to the forensic operational reality.

Author Response

Response to Reviewer #1

We are grateful for your helpful comments regarding our manuscript. Our responses to your comments are shown below.

Comments:

The paper is linked, as the authors write, to verifying changes in bone exposed to various environmental conditions over long periods of time. The authors estimating the PMI of skeletal remains based on analyzes of DNA quantity, element concentrations, and bone density in one bovine compact bones, each exposed to one of five environmental conditions (ie, seawater, freshwater, underground, outdoor, and indoor conditions) for 1 year.

The paper has seen an improvement work compared to the previous version and with further modifications it may become acceptable.

Response:

Thank you very much for kind comments.

Comments:

The major limitations that still persist in the study are related to the lack of description of the objective limits of this experiment.

Response:

We agree with the comments. According to the indication, we revised the sentences (lines 318-323) in Discussion as follows,

“We admit that this study is a limited experimental research and not applicable to forensic cases as it is. Particularly, we removed soft tissues prior to environmental exposure, which is not realistic in actual cases. Moreover, we did not perform a blind PMI evaluation according to the data obtained in this study. However, we might be said that this study is valuable for identifying the changes in bone after skeletonization. Further studies using more realistic specimens would bring more informative results for estimating the PMI. Regarding the result that bone density remained nearly unchanged in this study, it should be re-investigated using pig or dog femurs, which are more similar to those of human.” (lines 322-329, Highlighted in yellow)

Comments:

Furthermore, the minimum procedures for evaluating the results have not been respected, which should include an accurate methodology explaining who does what and on which scientific basis the estimation of the differences is reached.

Finally, I did not understand which PMI interval range is transferable from the experimental study to the forensic operational reality.

Response:

We apologize to lack of explanation and for hard to understand.

As shown in Fig.2, the result of 404bp showed a significant decrease of DNA quantities in seawater, freshwater, and underground at 3 days-1 week and in outdoor at 3 months. However, as shown in Fig.4, the difference in amplification of the two PCR targets (404bp and 128bp) showed a significant difference in seawater, underground, and outdoor at 1-3 months, indicating that the real-time PCRs targeting differentially sized targets could be an effective index for DNA degradation after 1-3 months. On the other hand, as shown in Fig.6, Na showed a significant decrease in freshwater at 6 months, and Ba showed a significant decrease in seawater and freshwater at 6 months. Also, Ba showed a significant difference in underground between 6 months and 1 year. However, additional analyses with more time points are needed to clarify the relation between the elements and the PMI.

Therefore, we revised the sentences (lines 313-316 in the previous manuscript) as follows,

“In the present study, we investigated various parameters for estimating the PMI of skeletal remains. We confirmed the usefulness of real-time PCR for determining the level of DNA degradation. Particularly, the difference in amplification of the two PCR targets (404 bp and 128 bp) may be an effective for estimating the PMI after 1-3 months. On the other hand, element concentrations determined by ICP-OES, particularly the concentrations of Na and Ba, changed with respect to environmental conditions and time. However, additional analyses with more time points are needed to clarify the relation between the elements and the PMI.” (lines 315-321 in the revised manuscript, Highlighted in yellow)

Reviewer 2 Report

I have reviewed this study a few weeks ago: undoubtedly, this research article is sensitively improved. Following the reviewers' suggestion, the authors have fixed different criticisms.
Nevertheless, I believe that several modifications should be made in the discussion section. Particularly, the authors should improve the limitation section, treating it as a separate paragraph (lines 318-323). Moreover, all limits should be inserted. Particularly, the authors aim to analyze the PMI, while they analyze the post mortem modifications after skeletonization! It remains a crucial point of the study: removing soft tissues represents an unrealistic situation. This important aspect should be discussed by the authors! In real cases, the time to lose all tissue during the post mortem modification should be considered. Moreover, the authors did not perform an evaluation of PMI: they observed the changes that occurred in the bone sample under different stored conditions and after different times. Finally, the authors did not evaluate the bones in blind conditions.

Author Response

Response to Reviewer #2

We are grateful for your helpful comments regarding our manuscript. Our responses to your comments are shown below.

Comments:

I have reviewed this study a few weeks ago: undoubtedly, this research article is sensitively improved. Following the reviewers' suggestion, the authors have fixed different criticisms.

Response:

Thank you very much for kind comments.

Comments:

Nevertheless, I believe that several modifications should be made in the discussion section.

Particularly, the authors should improve the limitation section, treating it as a separate paragraph (lines 318-323). Moreover, all limits should be inserted. Particularly, the authors aim to analyze the PMI, while they analyze the post mortem modifications after skeletonization! It remains a crucial point of the study: removing soft tissues represents an unrealistic situation. This important aspect should be discussed by the authors! In real cases, the time to lose all tissue during the post mortem modification should be considered. Moreover, the authors did not perform an evaluation of PMI: they observed the changes that occurred in the bone sample under different stored conditions and after different times. Finally, the authors did not evaluate the bones in blind conditions.

Response:

We agree with the comment. According to the indication, we treated the limitation section as a separate paragraph and revised it as follows,

“We admit that this study is a limited experimental research and not applicable to forensic cases as it is. Particularly, we removed soft tissues prior to environmental exposure, which is not realistic in actual cases. Moreover, we did not perform a blind PMI evaluation according to the data obtained in this study. However, we might be said that this study is valuable for identifying the changes in bone after skeletonization. Further studies using more realistic specimens would bring more informative results for estimating the PMI. Regarding the result that bone density remained nearly unchanged in this study, it should be re-investigated using pig or dog femurs, which are more similar to those of human.” (lines 322-329 in the revised manuscript, Highlighted in yellow)

Reviewer 3 Report

The manuscript has not changed substantially but the authors have slightly improved the discussion and added new references. It shows very preliminary results, however, I think this study deserves to be published as it could be useful for future studies as starting point or for comparison of results.  

I would suggest some changes that could improve it even more.

Line 24: I would specify in the abstract that the environmental conditions are not real, but simulated. I think It would give a better idea of what the reader will find in the paper. With the current description it seems that the study has been carried out under real conditions, and the reader may be disappointed when, on reading the whole article, he or she finds that these conditions were artificially simulated.

Line 138: I was unable to find out by “Promega KK” which control DNA was used for rtPCR. Maybe it is discontinued. Could the authors check if the name is correct and add a more precise reference to identify it?

Lines 197 to 199: The newly added sentence may have a small typo. I believe that, on line 198, “among in 4 conditions” may be “among the 4 conditions”. Otherwise, I would not understand the sentence. If the authors wanted to write something different, please, explain or clarify it a little more.

Lines 234 to 236: I think it does not make much sense to compare the quantification values obtained by spectrophotometry and by rtPCR. Spectrophotometry values will always (or in most cases) be higher, as it measures all types of DNA, including exogenous and damaged DNA; and rtPCR will only quantify the mtDNA of bovine origin that is in good condition to be amplified.

Author Response

Response to Reviewer #3

We are grateful for your helpful comments regarding our manuscript. Our responses to your comments are shown below.

Comments:

The manuscript has not changed substantially but the authors have slightly improved the discussion and added new references. It shows very preliminary results, however, I think this study deserves to be published as it could be useful for future studies as starting point or for comparison of results. 

I would suggest some changes that could improve it even more.

Response:

Thank you very much for kind comments.

Comments:

Line 24: I would specify in the abstract that the environmental conditions are not real, but simulated. I think It would give a better idea of what the reader will find in the paper. With the current description it seems that the study has been carried out under real conditions, and the reader may be disappointed when, on reading the whole article, he or she finds that these conditions were artificially simulated.

Response:

We thank you for nice advising. According to the comment, we added the word “simulated” to the sentence of line 24 as follows,

“simulated environmental conditions (seawater, freshwater, underground, outdoor, and indoor)” (Highlighted in yellow)

Comments:

Line 138: I was unable to find out by “Promega KK” which control DNA was used for rtPCR. Maybe it is discontinued. Could the authors check if the name is correct and add a more precise reference to identify it?

Response:

We apologize for uncertain information. We added the following information,

“Code No. MO-COW 15000/Lot No. PCW180818“ (Highlighted in yellow)

Comments:

Lines 197 to 199: The newly added sentence may have a small typo. I believe that, on line 198, “among in 4 conditions” may be “among the 4 conditions”. Otherwise, I would not understand the sentence. If the authors wanted to write something different, please, explain or clarify it a little more.

Response:

We apologize the mistake. We deleted the word “in” as follows,

“among the 4 conditions” (Highlighted in yellow)

Comments:

Lines 234 to 236: I think it does not make much sense to compare the quantification values obtained by spectrophotometry and by rtPCR. Spectrophotometry values will always (or in most cases) be higher, as it measures all types of DNA, including exogenous and damaged DNA; and rtPCR will only quantify the mtDNA of bovine origin that is in good condition to be amplified.

Response:

According to the comments, we revised the sentences (lines 290-299 in the previous manuscript) as follows,

“Total DNA varied in amount with conditions and exposure periods. In this quantitation, exogenous DNA originated from microorganisms proliferating in bone and endogenous DNA of bone itself are quantitated at once regardless of the degree of DNA degradation. Exogenous DNA begins to increase in parallel with severe bone degradation over a long exposure period. The values of total DNA, therefore depend on the valance between the amount of endogenous and exogenous DNA. Varieties in increase and decrease observed in this short termed study must be derived from this characteristic of quantitation. However, if an increase in total DNA is clearly observed in more highly degraded specimens, this quantitation can be used as an index for bone degradation.” (lines 291-299 in the revised manuscript, Highlighted in yellow)

Round 2

Reviewer 1 Report

The paper has been improved.

I am satisfied.

Author Response

Response to Reviewer #1

Reviewer 2 Report

The authors have further improved the manuscript. Nevertheless, they should fix a few minor points before the publication.
1. Considering the study's limitations, I suggest modifying the title in "Evaluation of Parameters for Estimating the Postmortem Interval of Skeletal Remains Using Bovine Femurs. A pilot study".
2. In the "abstract" section, I suggest inserting a short phrase about the study's limitations.
3. Lines 290-299: the authors have discussed the influence of the insects in the post mortem changes on human bones, particularly in the DNA quality, without any reference. Please, insert the suggested reference in the previous revision: - "First report of Heleomyzidae (Diptera) recovered from the inner cavity of an intact human femur" DOI: 10.1016/j.jflm.2019.05.021.

Author Response

Response to Reviewer #2

We are grateful for your helpful comments regarding our manuscript. Our responses to your comments are shown below.

Comments:

The authors have further improved the manuscript. Nevertheless, they should fix a few minor points before the publication.

  1. Considering the study's limitations, I suggest modifying the title in "Evaluation of Parameters for Estimating the Postmortem Interval of Skeletal Remains Using Bovine Femurs. A pilot study".
  2. In the "abstract" section, I suggest inserting a short phrase about the study's limitations.
  3. Lines 290-299: the authors have discussed the influence of the insects in the post mortem changes on human bones, particularly in the DNA quality, without any reference. Please, insert the suggested reference in the previous revision: - "First report of Heleomyzidae (Diptera) recovered from the inner cavity of an intact human femur" DOI: 10.1016/j.jflm.2019.05.021.

Response:

We agree with all the comments.

  1. According to the indication, we added the words “A pilot study” to the Title as follows,

“Evaluation of Parameters for Estimating the Postmortem Interval of Skeletal Remains Using Bovine Femurs: A Pilot Study” (Highlighted in yellow)

  1. Also, we inserted a phrase about the study’s limitations in the Abstraction section as follows,

“However, this study is a limited experimental research and not applicable to forensic cases as it is” (lines 33-34 in the re-revised manuscript, Highlighted in yellow)

  1. Similarly, according to the comment, we reinserted the previous sentences in the Discussion as follows,

“In addition, Sessa et al. [27] reported the influence of the insects in postmortem changes on bones. They mentioned that the presence of insects feeding on the marrow, could be one of the reasons of the poor DNA quality, however, other factors such as the environmental conditions where the skeleton was found cannot be excluded. The results may support ours.” (lines 300-304 in the re-revised manuscript, Highlighted in yellow)

This manuscript is a resubmission of an earlier submission. The following is a list of the peer review reports and author responses from that submission.

Round 1

Reviewer 1 Report

I found this paper very disappointing as it is an extremely limited experimental research and not very useful for forensic application purposes and equally confused in the methodology of approach followed by the authors.
The authors present the research as a preliminary study, in which they investigated changes in DNA quantity, elemental composition, and bone density in bovine femurs experimentally exposed to five conditions (seawater, freshwater, underground, outdoor, and indoor) for a maximum of 1 year.
In truth, they study a bovine femur and place the bone fragments in different study conditions to reach even preliminary conclusions but absolutely not comparable and of very limited scientific interest.
As the authors themselves admit, "However, further investigations are required to verify the effectiveness of this approach. Because bone density remained nearly unchanged in the bovine femur, it should be re-investigated using pig or dog femurs, which are more similar to those of humans. This preliminary study provides a basis for further studies of bones from humans or closely related species in a wider range of natural conditions over long time periods ".
The paper is difficult to read and too much data is offered on which, under discussion, the results are repeated without application ideas.
In fact, the discussion is a literature report on papers that are too different from the study conditions proposed by the authors.
I find that the discussion is repetitive with respect to the results previously presented by the authors and of no practical use (this is the major limitation of the research that is offered to readers).

Author Response

Response to Reviewer #1

We are grateful for your valuable comments. Our responses to your comments are shown below.

Comments:

I found this paper very disappointing as it is an extremely limited experimental research and not very useful for forensic application purposes and equally confused in the methodology of approach followed by the authors.

The authors present the research as a preliminary study, in which they investigated changes in DNA quantity, elemental composition, and bone density in bovine femurs experimentally exposed to five conditions (seawater, freshwater, underground, outdoor, and indoor) for a maximum of 1 year.

In truth, they study a bovine femur and place the bone fragments in different study conditions to reach even preliminary conclusions but absolutely not comparable and of very limited scientific interest.

As the authors themselves admit, "However, further investigations are required to verify the effectiveness of this approach. Because bone density remained nearly unchanged in the bovine femur, it should be re-investigated using pig or dog femurs, which are more similar to those of humans. This preliminary study provides a basis for further studies of bones from humans or closely related species in a wider range of natural conditions over long time periods ".

The paper is difficult to read and too much data is offered on which, under discussion, the results are repeated without application ideas.

In fact, the discussion is a literature report on papers that are too different from the study conditions proposed by the authors.

I find that the discussion is repetitive with respect to the results previously presented by the authors and of no practical use (this is the major limitation of the research that is offered to readers).

Response:

Thank you for the comments. We agree with the indication that this research has major limitation. However, we believe that the paper could give researchers some valuable information. Also, if possible, we would like to report the results of this study with even a small amount of information. To emphasize the limitations of this research, we revised the sentences in the Discussions as follows,

“However, we admit that this research has a major limitation. Further investigations are required to verify the effectiveness of this approach. Because bone density remained nearly unchanged in the bovine femur, it should be re-investigated using pig or dog femurs, which are more similar to those of humans. In addition, further studies of bones from humans or closely related species are required in a wider range of natural conditions over long time periods.“

Reviewer 2 Report

This study aims to evaluate the changes in the amount of DNA, element concentrations, and bone density in bovine femurs stored under different conditions: seawater, freshwater, underground, outdoor, and indoor. Moreover, each parameter has been evaluated at different times: after 3 days, 1 week, 1 month, 3 months, 6 months, and 1 year.

This pilot study could be very useful in the evaluation of the PMI in real forensic cases. Nevertheless, several limitations should be indicated. First of all, the bone used has been worked without any tissue: this is unrealistic. In real cases, the time to lose all tissue during the post mortem modification should be considered. Moreover, the authors did not perform an evaluation of PMI: they observed the changes that occurred in the bone under different conditions. In other words, the authors did not evaluate the bones in blind conditions, but they described the findings related to different PMI and different environmental conditions.

The abstract section should be improved. Particularly I suggest changing the study's aims. The authors investigated the changes that occurred in the bone samples after different times and under different environmental conditions.

The introduction section should be shortened, erasing the unnecessary information. Moreover, as previously indicated, the study's aims should be improved.

The Materials and Methods section described the experimental model, allowing the repetition.

The results section should be improved. Particularly, in section 3.2 (DNA quantitation of Bones Exposed to Various Environmental Conditions), the DNA quantitation of a 404-bp target should be summarized through a box plot analysis.

The Discussion section should be improved. Particularly, the authors missed indicating the study limitations. In this experimental model several limitations should be described:
1 - removing soft tissues, the authors have evaluated the post mortem modifications after skeletonization, no the PMI!
2 - sampling the bone, the authors have modified the original composition of the bone, exposing the internal zone to the postmortem modification;
3 - the authors have missed discussing the influence of the insects in the post mortem changes on human bones, particularly in the DNA quality. In this regards, you can read: "First report of Heleomyzidae (Diptera) recovered from the inner cavity of an intact human femur" DOI: 10.1016/j.jflm.2019.05.021.

Author Response

Response to Reviewer #2

We are grateful for your helpful comments regarding our manuscript. Our responses to your comments are shown below.

Comments:

This study aims to evaluate the changes in the amount of DNA, element concentrations, and bone density in bovine femurs stored under different conditions: seawater, freshwater, underground, outdoor, and indoor. Moreover, each parameter has been evaluated at different times: after 3 days, 1 week, 1 month, 3 months, 6 months, and 1 year.

This pilot study could be very useful in the evaluation of the PMI in real forensic cases. Nevertheless, several limitations should be indicated. First of all, the bone used has been worked without any tissue: this is unrealistic. In real cases, the time to lose all tissue during the post mortem modification should be considered. Moreover, the authors did not perform an evaluation of PMI: they observed the changes that occurred in the bone under different conditions. In other words, the authors did not evaluate the bones in blind conditions, but they described the findings related to different PMI and different environmental conditions.

Response: 

As the reviewer indicated, we used only bone without soft tissues. We agree with the comment that this is unrealistic experiment. We thought to observe only the changes of bone as a pilot study because more factors of soft tissues should affect the PMI, and to search novel something factors under skeletal remains.

Comments:

The abstract section should be improved. Particularly I suggest changing the study's aims. The authors investigated the changes that occurred in the bone samples after different times and under different environmental conditions.

Response:

According to the comments, we revised the study’s aims in the abstract as follows,

“To identify effective parameters for estimating the PMI of skeletal remains, we investigated the change of bone focused on the amount of DNA, element concentrations, and bone density that occurred in the bone samples of bovine femurs, each maintained under one of five environmental conditions (seawater, freshwater, underground, outdoor, and indoor) for 1 year.”

Comments:

The introduction section should be shortened, erasing the unnecessary information. Moreover, as previously indicated, the study's aims should be improved.

Response:

According to the comment, we shortened the contents of the Introduction by erasing the following sentences,

“In the early phase of postmortem changes, the progression of hypostasis, rigor mortis, and a decrease in body temperature are generally used to estimate the PMI. However, in the next phase, the estimation of the PMI becomes difficult as body decomposition proceeds. Several markers used in botanical and entomological investigations could be effective for the estimation of the PMI [1–3]. For example, accumulated-degree-days is a parameter that accounts for the different stages of body decomposition [4,5]. Furthermore, the longer the PMI, the more difficult it is to determine the time of death [6–8].”

Also, the last sentences in the Introduction was revised as follows,

“Therefore, in this study, the change of bone to find effective parameters for estimating the PMI of skeletal remains were investigated based on analyses of DNA quantity, element concentrations, and bone density in bovine compact bones, each exposed to one of five environmental conditions (i.e., seawater, freshwater, underground, outdoor, and indoor conditions) for 1 year.”

Comments:

The results section should be improved. Particularly, in section 3.2 (DNA quantitation of Bones Exposed to Various Environmental Conditions), the DNA quantitation of a 404-bp target should be summarized through a box plot analysis.

Response:

According to the comment, we summarized the data of section 2.2 through a box plot analysis (Figure 2-5). Also, we added Dunnett’s test as a statistical analysis in the Materials and Methods.

Comments:

The Discussion section should be improved. Particularly, the authors missed indicating the study limitations. In this experimental model several limitations should be described:

1 - removing soft tissues, the authors have evaluated the post mortem modifications after skeletonization, no the PMI!

2 - sampling the bone, the authors have modified the original composition of the bone, exposing the internal zone to the postmortem modification;

3 - the authors have missed discussing the influence of the insects in the post mortem changes on human bones, particularly in the DNA quality. In this regards, you can read: "First report of Heleomyzidae (Diptera) recovered from the inner cavity of an intact human femur" DOI: 10.1016/j.jflm.2019.05.021.

Response:

We agree with the comments. According to the indication, we revised the sentences in the Discussion as follows,

“we evaluated the postmortem modifications after skeletonization based on removing soft tissues and exposing the internal zone of bone.”

Reviewer 3 Report

The manuscript entitled “Evaluation of Parameters for Estimating the Postmortem Interval of Skeletal Remains Using Bovine Femurs” describes a study conducted to find biological, chemical or physical parameters in bones that vary over time, in order to define new markers useful for dating the postmortem interval (PMI) of skeletal remains. This is interesting for forensic sciences and it can be a helpful article that serves as a starting point for other similar studies that want to go further in the search for markers that allow the estimation of PMI in skeletal remains. Therefore, I believe that this work deserves to be published, although I would suggest making several modifications beforehand.

1. The abstract shows some results that cannot be found in the manuscript. In lines 30-31 it says there is significant decrease in bone density for seawater sample, but in the results shown in the manuscript there is none difference in bone density for any of the conditions studied.

2. The abstract seems too ambitious when it says that the parameters studied in this work are useful indexes for estimating the PMI (lines 31-34). I think that further analyses would be required to affirm this. Thus, I would suggest to say that these parameters may or could be useful indexes.

3. The numeration of the references cited along the text should be reviewed, since there are some inconsistencies, such as those in line 60: Walden et al. is reference 11, not 8; Gallelo et al. is reference 25, not 26. Thus, all the cites and references should be checked.

4. In Materials and Methods it is not clear if the specimens of the study were analysed immediately after they were retired from the investigated conditions, or if they were kept frozen until the end of the experiment (or even more time) and then analysed all together. In the same way, it is not clear when was the control specimen analysed, and if it was conserved also frozen until the end of the experiment. This should be clarified in this section.

5. I would suggest to change the title 2.3 “DNA extraction and PCR” for “DNA extraction and quantification”, because what is explained here are both types of quantifications, not only the one base on rt-PCR, but also the spectrophotometric one.

6. Also, in this section, would be nice to indicate the total volume of the PCR reactions and the amount or volume of DNA added to each reaction. The volume of the DNA extract added to the PCR reaction may be determinant when analysing the effect of inhibitory substances coextracted with the DNA in the PCR.

7. The visual observation of bone slices showed that those that were underground had different coloration, one with dark brown stains and the other one without them. I would suggest to clarify which of these specimens was used for DNA analyses and which one for the chemical and physical tests.

8. After DNA quantification by rt-PCR the authors draw different conclusions from, what I think are, the same results. In line 194 they say that specimens from indoor conditions do not show a DNA quantity decrease over the time, in contrast to the other conditions. However, when the authors describe the results showed in fig. 3 (lines 212-213), they claim that all environmental conditions showed a decreasing trend after 1 year. I agree with this last statement, since, while specimens of indoor condition seem to be more stable during the first 6 months, the amount of DNA obtained after 1 year drops in comparison to the control specimen. Thus, there is a decrease.

This incongruent conclusion is also repeated in “Discussion” (lines 272-273). It is contradictory, repeated throughout the manuscript, and it makes it confusing.

On another hand, the DNA amount quantified for indoor condition after 1 year is marked as statistically significant when compared to the control in figure 3, but it is not marked as significant in figure 2. Since I think that data used to draw both graphs are the same, I believe the conclusion in line 194 and the graphs are wrong. However, I might have misunderstood it. Please, clarify this point.

9. Again, when comparing fig. 2 and fig. 3, I have found some discrepancies that raise questions for me.

In figure 2, none of the outdoor measures are significantly different compared to the control? Why in this graph measures for 1 year of outdoor and indoor conditions are not significant (they don't have any "*" mark) when comparing to the control, but they are significant in next graph (fig. 3)?

10. I think there might be a mix-up between spectrophotometry and spectrofluorometry. I believe the method used for total DNA quantification is based in spectrophotometry. However, in lines 228 and 233 it says spectrofluorometry.

11. A preliminary test was performed for elementary analysis by ICP-OES. Even if it is explained in “Materials and Methods”, I would suggest to say in “Results” (lines 240-241) again that this test was done with samples of the specimens recovered after 6 months.

12. In lines (247-249) it says “In underground conditions, there was a significant difference in the Ba content between 6 months and 1 year (p < 0.01)”. Here the significance level showed is 0.01, however, in fig. 5 the columns for underground at 6 months and 1 year are marked with only one “*”, what is defined in the description of the figure as p < 0.05. What is the correct significance value?

On the other hand, did underground specimen of 1 year show differences with the control? It seems to have similar values to seawater and freshwater specimens, which have significant differences compared to the control, but it doesn't have any "*" in the graph.

Could the authors check these points and correct them if necessary?

13. When the results of bone density analysis are described, the authors say that densities tended to decrease, even though they are no statistically significant changes. I think it is not correct to say this. It's true that in most of the conditions, except for outdoor, samples of 1 year have slightly lower average density. But as the authors said, it is not statistically significant. Furthermore, densities at 6 months are lower than at 1 year in all the conditions. Thus, I do not think there is a decreasing tendency for bone density.

14. In “Discussion” (line 275) it is said that PCR inhibitors can affect the DNA quantity. This is not completely true. I would suggest to modify it and specify that PCR inhibitors would affect the results obtained by real-time PCR quantification, not “DNA quantity” in general. They do not have to affect necessarily other quantification methods, such as spectrophotometry or spectrofluorometry.

15. In lines 288-289, the authors should specify if the significant difference found in PCR amplification of both targets at 1 month refers to all the conditions tested or only to some of them. Otherwise, it seems that it applies to all the conditions, and this is not true.

16. I would suggest to better explain the first part of the sentence in lines 289-291 because I am not sure about what “this trend” refers to exactly.

In fig. 4, specimens for 1 year do not have significant differences between 128 and 404 bp targets, or, at least, they are not marked with “*”. 

Specimens of 1 year from indoor and outdoor conditions seem to be similar for both targets. If indoor specimen does not have differences, it is difficult to believe that outdoor specimen of 1 year does, since the mean and SD seem to be really similar between targets. In any case, this is difficult to say seeing only a graph, so I would like the authors to check and clarify it.

If it refers to have a greater degradation along time, I think indoor has also this trend, but less strong than the other conditions.

17. In lines 325-326, the authors suggest to use pig or dog femurs in following studies. In the abstract they propose to use human bones. I would include these human bones also here, in the discussion.

18. Line 330. The description of the last figure does not include its number (figure 7). It should be added.

Author Response

Response to Reviewer #3

We are grateful for your helpful comments regarding our manuscript. Our responses to your comments are shown below.

Comments:

The manuscript entitled “Evaluation of Parameters for Estimating the Postmortem Interval of Skeletal Remains Using Bovine Femurs” describes a study conducted to find biological, chemical or physical parameters in bones that vary over time, in order to define new markers useful for dating the postmortem interval (PMI) of skeletal remains. This is interesting for forensic sciences and it can be a helpful article that serves as a starting point for other similar studies that want to go further in the search for markers that allow the estimation of PMI in skeletal remains. Therefore, I believe that this work deserves to be published, although I would suggest making several modifications beforehand.

Response:

Thank you very much for kind comments.

Comments:

  1. The abstract shows some results that cannot be found in the manuscript. In lines 30-31 it says there is significant decrease in bone density for seawater sample, but in the results shown in the manuscript there is none difference in bone density for any of the conditions studied.

Response:

We apologize to leave the wrong sentence. We deleted the following sentence,

“Bone density measured using micro X-ray computed tomography decreased significantly (p < 0.01) in the sample immersed in seawater between 6 months and 1 year.”

Comments:

  1. The abstract seems too ambitious when it says that the parameters studied in this work are useful indexes for estimating the PMI (lines 31-34). I think that further analyses would be required to affirm this. Thus, I would suggest to say that these parameters may or could be useful indexes.

Response:

We agree with the comments. According to the indication, we revised the sentences as follows,

“This preliminary study suggests that the level of DNA degradation determined by real-time polymerase chain reaction and element concentrations determined by inductively coupled plasma optical emission may be useful indexes for estimating the PMI of victims under a wide range of environmental conditions.”

Comments:

  1. The numeration of the references cited along the text should be reviewed, since there are some inconsistencies, such as those in line 60: Walden et al. is reference 11, not 8; Gallelo et al. is reference 25, not 26. Thus, all the cites and references should be checked.

Response:

We apologize the mistakes. We checked all references and corrected them.

Comments:

  1. In Materials and Methods it is not clear if the specimens of the study were analysed immediately after they were retired from the investigated conditions, or if they were kept frozen until the end of the experiment (or even more time) and then analysed all together. In the same way, it is not clear when was the control specimen analysed, and if it was conserved also frozen until the end of the experiment. This should be clarified in this section.

Response:

We apologize to lack of explanation. We added the following sentence in the Materials and Methods,

“The specimens were analyzed almost in a week after they were retired from the investigated conditions. Control was analyzed at the same time of 3 days conditions.”

Comments:

  1. I would suggest to change the title 2.3 “DNA extraction and PCR” for “DNA extraction and quantification”, because what is explained here are both types of quantifications, not only the one base on rt-PCR, but also the spectrophotometric one.

Response:

We apologize the inappropriate title. We corrected the title as follows,

“DNA extraction and Quantification”

Comments:

  1. Also, in this section, would be nice to indicate the total volume of the PCR reactions and the amount or volume of DNA added to each reaction. The volume of the DNA extract added to the PCR reaction may be determinant when analysing the effect of inhibitory substances coextracted with the DNA in the PCR.

Response:

According to comment, we added the following sentence,

“Total volume of PCR reaction was 25 µL and the template DNA was 2 µL.”

Comments:

  1. The visual observation of bone slices showed that those that were underground had different coloration, one with dark brown stains and the other one without them. I would suggest to clarify which of these specimens was used for DNA analyses and which one for the chemical and physical tests.

Response:

These specimens were used for each experiment in spite of coloration.

Comments:

  1. After DNA quantification by rt-PCR the authors draw different conclusions from, what I think are, the same results. In line 194 they say that specimens from indoor conditions do not show a DNA quantity decrease over the time, in contrast to the other conditions. However, when the authors describe the results showed in fig. 3 (lines 212-213), they claim that all environmental conditions showed a decreasing trend after 1 year. I agree with this last statement, since, while specimens of indoor condition seem to be more stable during the first 6 months, the amount of DNA obtained after 1 year drops in comparison to the control specimen. Thus, there is a decrease.

This incongruent conclusion is also repeated in “Discussion” (lines 272-273). It is contradictory, repeated throughout the manuscript, and it makes it confusing.

On another hand, the DNA amount quantified for indoor condition after 1 year is marked as statistically significant when compared to the control in figure 3, but it is not marked as significant in figure 2. Since I think that data used to draw both graphs are the same, I believe the conclusion in line 194 and the graphs are wrong. However, I might have misunderstood it. Please, clarify this point.

Response:

Thank you for your comments. In Figure 2, that was almost close but not statistically significant by Turkey-Kramer test between control and 1 year of Indoor condition. We re-tested Dunnett’ s test for the same data, but that was not quite statistically significant either.

Comments:

  1. Again, when comparing fig. 2 and fig. 3, I have found some discrepancies that raise questions for me.

In figure 2, none of the outdoor measures are significantly different compared to the control? Why in this graph measures for 1 year of outdoor and indoor conditions are not significant (they don't have any "*" mark) when comparing to the control, but they are significant in next graph (fig. 3)?

Response:

We thank you for nice advising. We tested 404 bp of all environmental conditions (Figure 2), and 404 bp of 1 month and 1 year (Figure 3) by using Turkey-Kramer test. The cause of difference between figure 2 and 3 was the difference of number of groups and there might be affected by the values of 1 year of seawater, freshwater, and underground which were very small amount. We decided to use Dunnett’ s test for 404 bp of all environmental conditions (Figure 2), and the test was also used for Figure 5.

Comments:

  1. I think there might be a mix-up between spectrophotometry and spectrofluorometry. I believe the method used for total DNA quantification is based in spectrophotometry. However, in lines 228 and 233 it says spectrofluorometry.

Response:

According to the comment, we changed the word “spectrofluorometry” to “spectrophotometry”.

Comments:

  1. A preliminary test was performed for elementary analysis by ICP-OES. Even if it is explained in “Materials and Methods”, I would suggest to say in “Results” (lines 240-241) again that this test was done with samples of the specimens recovered after 6 months.

Response:

According to the comments, we added the following sentence in the Results as follows,

“Analysed samples were the control, seawater, freshwater, and underground specimens after 6 months (n = 1).”

Comments:

  1. In lines (247-249) it says “In underground conditions, there was a significant difference in the Ba content between 6 months and 1 year (p < 0.01)”. Here the significance level showed is 0.01, however, in fig. 5 the columns for underground at 6 months and 1 year are marked with only one “*”, what is defined in the description of the figure as p < 0.05. What is the correct significance value?

Response:

We apologize the mistake. We corrected as 5 %.

On the other hand, did underground specimen of 1 year show differences with the control? It seems to have similar values to seawater and freshwater specimens, which have significant differences compared to the control, but it doesn't have any "*" in the graph.

Response:

There was almost but not quite statistically significant between the control and 1 year of underground.

Could the authors check these points and correct them if necessary?

Response:

We checked them.

Comments:

  1. When the results of bone density analysis are described, the authors say that densities tended to decrease, even though they are no statistically significant changes. I think it is not correct to say this. It's true that in most of the conditions, except for outdoor, samples of 1 year have slightly lower average density. But as the authors said, it is not statistically significant. Furthermore, densities at 6 months are lower than at 1 year in all the conditions. Thus, I do not think there is a decreasing tendency for bone density.

Response:

According to the comments, we revised the sentence as follows,

“Although the densities tended to decrease at 6 months and increase at 1 year, these changes were not statistically significant.”

Comments:

  1. In “Discussion” (line 275) it is said that PCR inhibitors can affect the DNA quantity. This is not completely true. I would suggest to modify it and specify that PCR inhibitors would affect the results obtained by real-time PCR quantification, not “DNA quantity” in general. They do not have to affect necessarily other quantification methods, such as spectrophotometry or spectrofluorometry.

Response:

We apologize the mistake. We revised the sentence as follows,

“In addition, PCR inhibitors in soil can affect the DNA quantity that measured by real-time PCR.”

Comments:

  1. In lines 288-289, the authors should specify if the significant difference found in PCR amplification of both targets at 1 month refers to all the conditions tested or only to some of them. Otherwise, it seems that it applies to all the conditions, and this is not true.

Response:

According to the comments, we revised the sentences as follows,

“PCR amplification was significantly lower for the 404 bp target than for the 128 bp target at 3 months in seawater, 1 month and 3 months in underground, and outdoor conditions (Figure 4). This trend of DNA degradation at 1month and 3 months suggests that our real-time PCRs targeting differently sized targets could be an effective index for DNA degradation of short terms. For more long terms, further investigations using fragments shorter than 128 bp may be useful.”

Comments:

  1. I would suggest to better explain the first part of the sentence in lines 289-291 because I am not sure about what “this trend” refers to exactly.

Response:

We mentioned in the response to comment 15.

Comments:

In fig. 4, specimens for 1 year do not have significant differences between 128 and 404 bp targets, or, at least, they are not marked with “*”.

Specimens of 1 year from indoor and outdoor conditions seem to be similar for both targets. If indoor specimen does not have differences, it is difficult to believe that outdoor specimen of 1 year does, since the mean and SD seem to be really similar between targets. In any case, this is difficult to say seeing only a graph, so I would like the authors to check and clarify it.

If it refers to have a greater degradation along time, I think indoor has also this trend, but less strong than the other conditions.

Response:

Thank you for your comments. In Figure 4, that was almost close but not statistically significant by Turkey-Kramer test between control and 1 year of Indoor condition. We re-tested Dunnett’ s test for the same data, but that was not quite statistically significant either.

Comments:

  1. In lines 325-326, the authors suggest to use pig or dog femurs in following studies. In the abstract they propose to use human bones. I would include these human bones also here, in the discussion.

Response:

According to the comments, we revised the sentence as follows,

“However, we admit that this research has a major limitation. Further investigations are required to verify the effectiveness of this approach. Because bone density remained nearly unchanged in the bovine femur, it should be re-investigated using pig or dog femurs, which are more similar to those of humans. In addition, further studies of bones from humans or closely related species are required in a wider range of natural conditions over long time periods.”

Comments:

  1. Line 330. The description of the last figure does not include its number (figure 7). It should be added.

Response:

We apologize the mistake. We added the number.

Round 2

Reviewer 1 Report

The paper has remained absolutely unchanged, so a single sentence at the end of the paper cannot change my opinion.

Authors studied a bovine femur and place the bone fragments in different study conditions to reach even preliminary conclusions but absolutely not comparable and of very limited scientific interest.

Reviewer 2 Report

Following the reviewers' suggestion, the authors have sensitively improved their paper. I believe that it could be published in this form.

Reviewer 3 Report

The amendments made to the article have improved its quality.

However, I still have one issue that I would like the authors to clarify.

The new results added to the manuscript (the boxplot and the statistical analysis) show now in figure 2 that the DNA quantity in outdoor conditions at 1 year is significantly different to the control. However, the sentence at lines 198 to 200 says that it did not decrease substantially. 

This might be describing the results showed by the figure of the previous version. If this is the case, it should be adapted to the new results. Otherwise, could the authors explain it more clearly?